# Novel Therapeutic Targets for Migraine

**DOI:** 10.3390/biomedicines11020569

**Published:** 2023-02-15

**Authors:** Areeba Nisar, Zubair Ahmed, Hsiangkuo Yuan

**Affiliations:** 1Jefferson Headache Center, Department of Neurology, Thomas Jefferson University, Philadelphia, PA 19107, USA; 2Center for Neurological Restoration, Neurological Institute, Cleveland Clinic, Cleveland, OH 44195, USA

**Keywords:** migraine, pituitary adenylate cyclase-activating polypeptide, Delta opioid receptor, adenosine, therapeutics

## Abstract

Migraine, a primary headache disorder involving a dysfunctional trigeminal vascular system, remains a major debilitating neurological condition impacting many patients’ quality of life. Despite the success of multiple new migraine therapies, not all patients achieve significant clinical benefits. The success of CGRP pathway-targeted therapy highlights the importance of translating the mechanistic understanding toward effective therapy. Ongoing research has identified multiple potential mechanisms in migraine signaling and nociception. In this narrative review, we discuss several potential emerging therapeutic targets, including pituitary adenylate cyclase-activating polypeptide (PACAP), adenosine, δ-opioid receptor (DOR), potassium channels, transient receptor potential ion channels (TRP), and acid-sensing ion channels (ASIC). A better understanding of these mechanisms facilitates the discovery of novel therapeutic targets and provides more treatment options for improved clinical care.

## 1. Introduction

Migraine is a complex neurovascular disorder affecting nearly 15% of the world’s population [1]. It is the second global cause (level III) of years lived with disability (YLD) after low back pain, with 44.6 million YLD in 2019 [2]. The underlying mechanism of migraine is related to the dysfunctional activation of the trigeminovascular system [3]. The trigeminovascular neurons, along with afferents from the upper cervical cord, innervate multiple pain-sensitive structures in the head and communicate with the trigeminal cervical complex (TCC). This activation of the trigeminal system results in neurogenic inflammation and increased sensitivity to pain leading to sustained headache, triggering the release of various neuropeptides, including calcitonin gene-related peptide (CGRP), substance P, and others that play a key role in pain signaling and modulation [3,4]. CGRP is a 37 amino-acid that is widely distributed in both neural and non-neural areas of the body. CGRP is the most potent microvascular vasodilator known, with a potency that is 10 times greater than prostaglandins and 10–100 times more potent than vasodilators such as acetylcholine and substances P [5]. In the trigeminal system, CGRP is the most abundant neuropeptide and is found in small, capsaicin-sensitive C fibers that follow cerebral and meningeal arteries and innervate tissues. When trigeminal nociceptive fibers are activated, they release CGRP, which binds to the CGRP receptor and the amylin1 receptor on neurons, nodes of Ranvier, glia cells, and blood vessel smooth muscle that causes vasodilation, trigeminal activation, and sustained allodynia. The discovery of CGRP led to the development of four monoclonal antibodies (mAbs) (erenumab, fremanezumab, galcanezumab, eptinezumab) and three small molecules (ubrogepant, rimegepant, and atogepant) approved by the US Food and Drug Administration (FDA) to date. The American Headache Society and the European Headache Federation recommend incorporating CGRP small molecule antagonists and CGRP pathway-tarted mAbs into the treatment plan, as they are proven safe and efficacious as acute and preventive treatments for migraine [6,7].

Despite the high prevalence of migraine, many migraine medications are often poorly tolerated. Adherence to preventive medicine decreases significantly by 30 days, with only 25% adherence at six months [8]. Newer FDA-approved agents, such as CGRP pathway-targeted therapies, are generally better tolerated than conventional oral medications [9]. Further research in migraine therapeutics remains essential to optimize patient care. Emerging research in migraine pathophysiology has expanded the horizon of treatment targets and led to the discovery of many novel therapeutic options. In this narrative review, we focus on several investigational therapeutic targets that have undergone clinical trials. These novel targets include pituitary adenylate cyclase-activating polypeptide (PACAP), adenosine, δ-opioid receptors (DOR), potassium channels, transient receptor potential ion channels (TRP), and acid-sensing ion channels (ASIC) (Figure 1). 

## 2. Pituitary Adenylate Cyclase-Activating Polypeptide

In the CNS, PACAP is found at a high density in many of the structures associated with migraine and nociception, from sensory nuclei to ganglia [10]. PACAP belongs to the secretin superfamily, which includes glucagon and vasoactive intestinal peptide (VIP). It is an adenylate cyclase stimulator found in pituitary cells and was isolated in the late 1980s from ovine hypothalamic tissues [11]. It binds to PAC1, VPAC_1,_ and VPAC_2_, which are G-protein coupled receptors. PACAP is broadly expressed in two bioactive forms, PACAP38 and PACAP27, with the former being more prevalent in mammals [11]. Internalization capacity and cAMP accumulation vary among PACAP receptors, leading to a biased agonism [12]. PACAP and VIP are parasympathetic co-transmitters and mediators of various auto/paracrine functions, including thermogenesis, circadian rhythms, tissue perfusion, and smooth muscle response [10]. 

PACAP may be involved in migraine through pathways independent of CGRP, nitric oxide, and potassium. The peripheral administration of PACAP induced a reduction in hyperalgesia, allodynia, and nocifensive behavior [13]. PACAP gene-deficient mice demonstrated impaired nocifensive response and reduced trigeminovascular system activation [14]. Interestingly, PACAP shows independent effects despite partially co-localizing with CGRP in trigeminal ganglion (TG). PACAP38 provoked cutaneous hypersensitivity and carotid artery dilation in *Ramp1* knockout mice, suggesting a pathway independent of CGRP [15]. In an animal model, CGRP and PACAP mAbs could not block PACAP and CGRP-induced light aversion, respectively [16]. PACAP38-induced hypersensitivity can be partially blocked by a potassium channel (K_ATP_) inhibitor (glibenclamide), which has previously been shown to inhibit glyceryl trinitrate (GTN)-induced hypersensitivity completely [15]. Another animal model showed that PACAP mAbs did not act on signaling pathways activated by GTN and levcromakalim (K_ATP_ channel agonist) [17]. Sumatriptan also had no influence on PACAP38-induced hypersensitivity [17]. These findings suggest that PACAP may be involved in migraine via a unique pathway independent of other common migraine inducers.

Several human studies also confirmed PACAP’s role in migraine. An elevated PACAP38 serum level was found during the ictal phase; its level was negatively correlated with migraine duration [18]. Higher levels of PACAP were seen in chronic migraine (CM) patients compared to patients with episodic migraine (EM) or healthy controls [19]. PACAP infusion provoked migraine-like attacks in migraine patients and healthy individuals with a marked and prolonged dilation of the middle meningeal artery (MMA) [20,21]. PACAP causes greater premonitory migraine symptoms such as neck stiffness and thirst in patients (48%) compared to CGRP (9%) [22]. Furthermore, in a randomized, double-blind, placebo-controlled trial, erenumab could not prevent PACAP38-induced migraine-like attacks in migraine patients (NCT02542605; data unpublished). PACAP is therefore a potential therapeutic target for migraine. 

Several PACAP pathway-targeted mAbs have been investigated clinically. ALD1910 binds selectively to PACAP38 and PACAP27 [23], whereas AMG301 targets the PAC1 receptor. No result has yet been published for ALD1910 (NCT04197349), which completed its phase I double-blinded, randomized, placebo-controlled trial in 2019. AMG301, in a phase IIa randomized double-blinded placebo-controlled trial (NCT03238781), it demonstrated no efficacy benefit over placebo in 343 randomized patients on monthly migraine days or migraine-specific medication use days [24]. The lack of efficacy might be due to the differences in the affinity of AMG301 to PAC1 receptors, its concentration achieved at the target, or perhaps the selective inhibition of the PAC1 receptor is insufficient to reduce migraine frequency [25]. Despite the lack of trial-based evidence, the potential of PACAP mAbs for migraine remains high, thus requiring more research on its isoforms and receptors. Two ongoing clinical trials are evaluating different PACAP38 mAbs, LY3451838 (NCT04498910) and Lu AG09222 (NCT05133323), for the prevention of treatment-resistant migraine. These studies will offer more insight into PACAP mAbs for migraine prevention.

## 3. Adenosine Receptor

Adenosine is a purinergic vasoactive amine neurotransmitter with four G protein-coupled receptors (A_1_, A_2A_, A_2B_, A_3_) expressed throughout the body. It is involved in various conditions such as sleep regulation, mood disorders, arousal, epilepsy, and headache [26]. Adenosine displays either nociceptive (Gs-coupled A_2a_ and A_2b_) or antinociceptive (Gi-coupled A_1_ and A_3_ receptors) effects depending on the receptor subtype [27]. Each receptor type is widely expressed throughout the central and peripheral nervous system, with an abundance in the expression of A_1_ and A_2a_ receptors in TG and trigeminal nucleus caudalis (TNC) [28]. Adenosine is thought to play an essential role in sleep induction; conversely, caffeine is a non-selective adenosine receptor antagonist [29]. Adenosine has been widely studied in the cardiovascular system, as it causes vasodilation through A_2A_ receptors, whereas in the atrioventricular and sinoatrial nodes, it binds with the A_1_ receptor, reducing heart rate. Adenosine may also play a role in migraine pathogenesis as a part of the breakdown product of adenosine triphosphate (ATP) and its binding to ATP-gated P2X3 receptors [30].

Adenosine is an attractive target for migraine, as it is associated with energy metabolism and the sleep-wake cycle. Human studies show increased adenosine blood levels ictally compared to interictally [31]. Dipyridamole, an adenosine reuptake inhibitor, induces headaches in healthy individuals and migraine patients [32]. Two recent systematic reviews of pre-clinical and clinical studies on the involvement of adenosine in migraine signaling pathways also support adenosine as a target due to its dual effect on vasodilation, expression of A_1_ and A_2a_ receptors in TG, and elevated ictal serum levels [31,33].

Several animal and human studies support the role of the A_1_ receptor as a target for migraine therapy. The A_1_ receptor agonist inhibits nociceptive signaling in peripheral pain models. As seen in animal studies of trigeminovascular nociception, selective A_1_ receptor activation inhibits neuronal activation without concomitant vasoconstriction [34]. This response of A_1_ receptors, coupled with a decrease in non-evoked pain behavior and evoked mechanical hyperalgesia after surgical paw incision, suggests the potential of the A_1_ receptor as a target for the pain management [35]. Another study shows that the activation of peripheral A_1_ receptors triggered an anti-nociceptive effect on glutamate-induced nociceptive behavior [36]. An A_1_ receptor agonist (GR79236) inhibited electronically induced vasodilation and CGRP release via a prejunctional inhibition [33]. In healthy human subjects, GR79236 inhibited trigeminal nociception as measured by the blink reflex [37]. However, to date, no A_1_ agonist has been approved for clinical use, probably due to its depressive cardiorespiratory effects. The discovery of a novel A_1_-selective agonist (benzyloxy-cyclopentyladenosine), which is a potent and powerful analgesic without cardiorespiratory suppression via biased agonism (selective activation of Gob without recruitment of β-arrestin) [38], may be a potential therapeutic molecule in the future. 

The blocking of the A_2A_ receptors, which are G stimulatory receptors that induce nociceptive behavior, may be another target for migraine. As seen in an animal model, electrical stimulation of TG upregulated both the A_2A_ receptors and CGRP [28]. A_2A_ receptor antagonists inhibit MMA vasodilation without modifying blood pressure [39,40]. The authors suggested that balanced action on A_1_ and A_2A_ receptors might be necessary to attenuate dural vasodilation completely [40]. 

Currently, there is only one ongoing clinical trial (NCT04577443) investigating adenosine-mediated effects on cranial hemodynamics and migraine. The primary outcome will compare the effects of adenosine vs. saline infusion in migraine and healthy patients 12 h after infusion. Secondary outcomes such as changes in headache frequency, cerebral hemodynamics, and arterial vasodilation will also be studied. Preclinical and clinical studies support selective A_1_ receptor activation and selective A_2A_ receptor inhibition as a potential drug target for migraine treatment. 

## 4. δ-Opioid Receptor

Opioid receptors (four classical types: μ, κ, δ, nociception/orphanin FQ) play an essential role in mediating various types of pain. However, most μ-opioid receptor (MOR) drugs on the market are unsuitable for treating migraine pain. They are well-known for causing tolerance, dependency, abuse, and medication overuse headache (MOH) while reducing the efficacy of concurrent migraine treatments such as triptans [41]. The biased MOR agonist, oliceridine, preferentially activates G protein signaling but less β-arrestin recruitment, thereby generating less respiratory depression and constipation than morphine [42]. In contrast to MOR, δ-opioid receptors (DOR) could be a more favorable target due to their analgesic, anxiolytic and antidepressant effects [43]. DORs are highly expressed throughout the brain, spinal cord, and primary sensory afferents. DORs within the dorsal root ganglia (DRG), TG, TNC, and forebrain structures are likely involved in processing pain and pain-related emotional and cognitive states, rendering DOR an attractive target for migraine, in which both anxiety and depression are co-morbid.

DOR agonists alter pain and allodynia in preclinical studies. A selective DOR agonist, SNC80, attenuated CGRP release and blocked pronociceptive CGRP signaling in an NTG model of chronic migraine. SNC80 also inhibited NTG-induced acute allodynia and the development of chronic allodynia [44], as well as evoked cortical spreading depression (CSD) [45]. This could be explained by the expression of DORs on the majority of dural CGRP fibers [46]. In addition, DOR agonists are effective in preclinical models of CM, MOH, opioid-induced hyperalgesia, and post-traumatic headache [45,47]. Another advantage seen in rodent models of SNC80 is the limited development of opioid-induced hyperalgesia (OIH), a condition seen in the chronic usage of opioids relative to sumatriptan or morphine [47]. DOR and CGRP are co-expressed in approximately 40% of total TG neurons. Based on animal study findings by Moye et al., it is speculated that DOR agonists may have their anti-nociceptive effect in two ways: by preventing CGRP release from TG neurons and potentially by preventing CGRP receptor pro-nociceptive signaling [44]. A limitation of DOR agonists is their propensity to reduce the seizure threshold. A non-convulsive DOR agonist, KNT-127, inhibited CSD and established cephalic allodynia without increasing the seizure potential [48]. These preclinical studies suggest that DOR agonists can be used as an acute and preventive migraine treatment.

Clinical data on the role of DOR agonists in migraine treatment and prevention is, however, more limited. A phase 1 clinical trial of a DOR agonist TRV250 (NCT04201080), injected subcutaneously in 38 healthy adults, found it well tolerated and supported the need for future phase 2 trials for acute migraine [49]. In a proof-of-concept RCT (NCT00759395), a selective DOR agonist AZD2327 demonstrated the potential of an anxiolytic effect (decrease vascular endothelial growth factor level) [50]; it may be explored in human studies of migraines with anxiety and depression. DOR agonists have the unique ability to attenuate pain, depression, and anxiety states while providing a potentially favorable adverse effect (AE) profile without the physical dependence seen with MORs.

## 5. Potassium Channels

K_ATP_ channels are from the family of inward-rectifying (Kir) transmembrane potassium channels. They are divided into seven subfamilies (Kir1.x to Kir7.x), with K_ATP_ channels belonging to the Kir6.x subfamily (subtypes; Kir6.1 and Kir6.2) and are associated with membrane electrophysiology and cellular metabolism. Functional Kir channels are composed of four Kir subunits and four sulphonylurea receptors (SUR2A, SUR2B, and SUR1). Kir6.1/SUR2B subunit combination has been suggested as a target for migraine as it is found in the vascular smooth muscles, dominantly in brain arteries, dura mater, and in TNC and TG [51,52,53]. The mRNA transcripts of SUR1 and Kir6.2 were predominantly found in the brain, pancreas, and heart [53]. K_ATP_ channels have neuroprotective effects in neurons during ischemia and oxidative stress [54]. Another potassium channel associated with migraines is the large (big)-conductance calcium-activated K^+^ channels (BK_Ca_) that are found in TG and TNC [55,56]. It is suggested that PACAP38 triggers the phosphorylation and activation of the K_ATP_ and BK_Ca_ channels [55].

K^+^ influx induces cerebral vasodilation and migraine, as seen in preclinical and clinical studies of K_ATP_ channel agonists. The K_ATP_ channel agonist (levcromakalim) is commonly used in CM rat models. The repeated administration of levcromakalim led to a significant increase in c-Fos expression in brain areas associated with migraine such as the medial prefrontal cortex and spinal trigeminal nucleus [57]. Headache and migraine-provoking substances have vasodilatory effects on the extracerebral and cerebral arteries. Levcromakalim induced an increase in cerebral blood flow and dilated the middle cerebral artery (MCA) in healthy volunteers [58]. Levcromakalim infusion caused migraine attacks mimicking spontaneous migraine in healthy volunteers, patients with migraine without aura (NCT03228355), and migraine with aura (NCT04012047) [59,60,61]. Levcromakalim dilated the superficial temporal artery and MMA but not MCA, probably due to its limited ability to cross the blood-brain barrier [59]. MaxiPost (BMS-204352), a BK_Ca_ channel opener, was found to initiate migraine attacks, increase MCA velocity, and dilate superficial temporal arteries and radial arteries [62]. Since Kir6.1 knocked-out showed no hypersensitivity in response to GTN and levcromakalim [51], Kir6.1 subunit is responsible for hypersensitivity response to K_ATP_ channel activation. Currently, there is an ongoing clinical trial (NCT05565001) assessing levcromakalim’s effect on meningeal nociceptors, ascending trigeminal nociceptive pathways, and CSD in migraine attacks (with and without aura). The results have yet to be announced, but they will provide more insight and understanding of K_ATP_ channels. 

With levcromakalim being able to induce migraine attacks, it is reasonable to investigate K_ATP_ channel blockers for migraine treatment. Glibenclamide, a sulfonylurea anti-diabetic drug that non-specifically blocks K_ATP_ channel opening, has been shown to inhibit spontaneous cephalic hypersensitivity and CGRP release in spontaneous allodynic rats [52]. CGRP antagonists (olegepant) also have inhibitory effects on levcromakalim-induced acute and basal hyperalgesia in CM rat models [57]. Glibenclamide and paxilline (BK_Ca_ channel inhibitor) successfully inhibited PACAP-induced cerebellar vascular dilation in animals, and a similar effect was reported in human pulmonary arteries [55]. Glibenclamide also induced a concentration-dependent constriction of isolated MMA [63]. Additionally, the K_ATP_ channel blocker (PNU-37883A) greatly reduced the in vivo dilatory effects of levcromakalim in MMA [64]. However, glibenclamide failed to attenuate headache and vascular changes induced by PACAP38 or CGRP infusions in healthy volunteers [65,66]. The failed efficacy of glibenclamide can be attributed to the low dose (10 mg highest tolerated) used in the study compared to the higher dosage used in animal studies (20–30 mg/kg) [52,65]. It is worth noting that glibenclamide has a higher affinity for the SUR1 subunit of non-specific K_ATP_ channels, which explains its non-efficacious effect on Kir6.2/SUR2B [67,68]. Kir6.1/SUR2B is the major functional K_ATP_ channel complex in MMA and MCA, whereas SUR2A is merely detected within the heart. The targeted blocking of Kir6.1 or SUR2B K_ATP_ channel subunits in large cerebral and meningeal arteries may be a future anti-migraine strategy as they might not have any CNS or cardiac or endocrine effect (insulin release) [53,64]. Using a SUR2B subunit-specific inhibitor might show higher efficacy in blocking the effect of PACAP38 due to a higher concentration of the SUR2B subunit in vascular smooth muscle cells. No selective SUR2B blockers are available for clinical use to date. K_ATP_ channels are a promising yet challenging target for researchers due to their ubiquitous distribution in the peripheral and CNS. Research in these channels and formulating specific targeted inhibitors could result in new drug options for migraine.

## 6. Transient Receptor Potential Ion Channels

TRP channels are a large family of cation channels with more than 50 members. They are activated by changes in the influx of ions like Ca^2+^ and Na^+^ leading to changes in cell membrane voltage and the activation of signaling pathways through secondary messenger systems. Out of the six TRP subfamilies, only TRP melastatin (eight members, TRPM1-8), TRP vanilloid (six members, TRPV1-6), and TRP ankyrin (one member, TRPA1) are associated with migraines. These channels, specifically TRPV1, TRPV2, TRPV3, TRPV4, TRPA1, and TRPM8, known as thermo-TRP channels, are sensitive to temperature changes. These subfamilies comprise the largest group of ion channels expressed in the sensory ganglion, particularly in the DRG, TG, and the vagal ganglia. Stimulating TRPV1, TRPV4, and TRPA1 results in the activation of neurogenic inflammation by releasing substance P (SP), CGRP, VIP, and PACAP (see reviews by Nilius et al. and Iannone et al.) [69,70]. The location of TRP channels, response to physical/chemical stimuli, and association with pain mediators suggest their role in migraine pathophysiology and as potential therapeutic targets. Here we discuss the TRP channels with more substantial preclinical evidence.

Preclinical studies show that the TRPV1 antagonistic mechanism is through its effect on CGRP. TRPV1 antagonists JNJ-38893777 and JNJ-17203212 decrease CGRP release in rodent models [71]. Other TRPV1 antagonists, A-993610 and SB-705498, failed to demonstrate any significant effects on migraine-like symptoms such as CSD [72,73]. TRPA1 is highly co-expressed with TRPV1 and is associated with familial pain syndrome and umbellulone (a naturally occurring migraine inducer) [74]. TRPA1 is also associated with CGRP release like TRPV1 [75]. TRPA1 is affected by other nociceptive agents as seen in spinal antinociceptive models of acetaminophen [76]. Studies also show that onabotulinumtoxinA can inhibit the expression of TRPA1 and TRPV1 [77]. Recently, a significant correlation has been found between the incidence of migraine and single nucleotide polymorphisms (SNPs) closely associated with the TRPM8 coding region [78,79]. The application of icilin (TRPM8 activator) in rats produces cutaneous allodynia, which is attenuated by pretreatments of TRPM8 antagonist (AMG1161) [79]. Furthermore, the behavioral response of cold stimulus on CGRP is also mediated by TRPM8 [80]. TRPM8 may be involved in the development of migraine as individuals with migraine protective alleles show reduced sensitivity to cold stimuli [81]. 

TRP channels are an attractive drug target for inflammatory pain conditions, but the alterations in thermoregulation may be a critical limitation. SB-705498 progressed to a Phase-II clinical trial (NCT00269022); however, the study was terminated on undisclosed grounds, and no results have been posted. TRPV1 and TRPA1 potentially play a role with CGRP in headaches and are also involved in sensitivity to environmental factors such as heat, pH changes, and natural migraine triggers. TRP channel-targeted drug development is a challenging process requiring considerable preclinical and clinical research due to the high sensitivity of TRP channels to many physiological and environmental stimuli in the body. 

## 7. Acid Sensing Ion Channels

ASIC are voltage-insensitive cation channels located primarily in the nervous system. ASIC1-4 isotypes are pH-sensitive, activated by lipids and low extracellular pH in brain ischemia and physiological synaptic transmission. Most isotypes are in TG, and ASIC1a and ASIC2 isotypes are in higher concentrations in DRG and TNC. 

ASIC inhibitors have shown potential therapeutic effects in migraine. As seen in preclinical models, they inhibit allodynia, CSD, and trigeminal activation. ASIC3 activators have been implicated in developing cutaneous allodynia and CSD [82]. Mambalgin-1, a specific inhibitor of ASIC1-containing channels, and amiloride, a non-specific ASICs blocker, reversed the development of facial allodynia and delayed chronic allodynia in animal models in a similar manner to sumatriptan and topiramate [83]. Reversible ASICs inhibitors have shown similar opioid-pathway independent analgesic effects involving ASC1a and ASC1b and a peripheral impact on the inflammation pathway involving ASIC1b [84]. Amiloride is also used as a diuretic and antihypertensive; however, the specifically targeted blockage of ASIC does not block epithelial sodium channels, which decreases the impact on blood pressure. Amiloride also inhibited CSD and trigeminal activation in animal models [83,85]. Amiloride inhibited the generation of sustained cephalic and extra cephalic allodynia, supporting the role of ASICs activation in migraine-related behaviors [82]. ASICs also affect other neuropeptides, as seen by a three-fold decrease in CGRP release from trigeminal neurons in the presence of ASIC3 inhibitors [86]. CGRP release also appears partially dependent on pH changes and may be affected by ASIC inhibition [86,87]. 

Human studies of ASIC are limited, and safety protocols need to be developed due to the non-specific effects of non-selective ASIC inhibitors such as amiloride. In a small open-label study of seven patients with refractory migraine and persistent aura, systemic treatment with amiloride reduced the frequencies of both aura and headache severity in 4/7 patients at a dose of 10–20 mg/day [85]. Limitations of ASIC targeting include the non-specific action of non-selective inhibitors and potential off-target AEs. A multicenter phase 2 clinical trial (NCT04063540) is currently evaluating the amiloride vs placebo therapeutic effects in patients with migraine with aura. Though preclinical data shows promising results on allodynia and CSD, more clinical trials are needed to establish the safety and efficacy of ASIC inhibitors in migraine patients.

## 8. Conclusions

Migraine is a complex neurological disease with a high burden of illness and functional impact on patients’ lives. The progress and development in science have uncovered the multifactorial aspect of migraine pathophysiology. This review has briefly discussed some current research targets in migraine medicine. Some insights from this review show an interlinkage of neuropeptides, such as CGRP, with PACAP and then PACAP with K_ATP_. These neuropeptides and their receptors need further exploration, animal studies, and these need to be followed with preclinical and clinical trials. The path opened due to CGRP discovery and usage provides the perfect foundation for neuropeptide and receptor-targeted drugs to be the future of migraine medication. A migraine therapeutic regime needs more medication options to personalize our patients’ treatment plans based on their needs and effects.

## Figures and Tables

**Figure 1 biomedicines-11-00569-f001:**
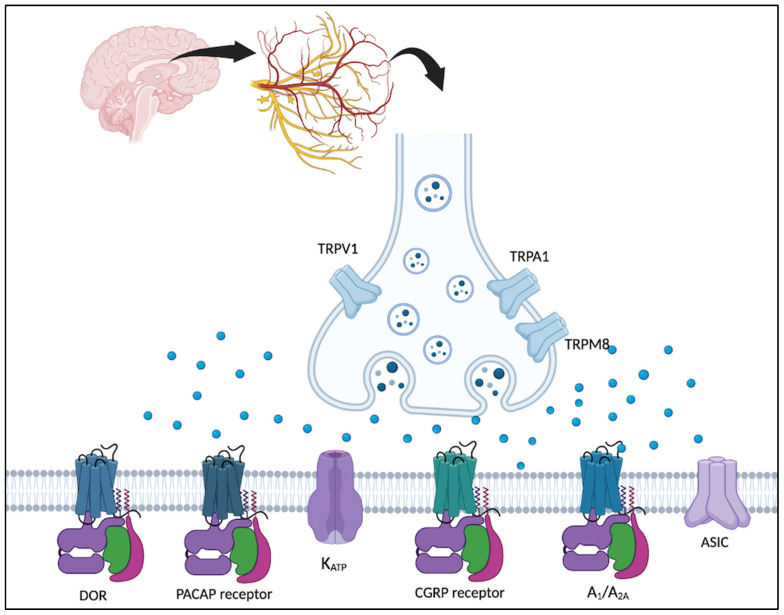
Illustration of several potential therapeutic targets under investigation for migraine management. Adenosine receptors (A_1_/A_2A_), Pituitary adenylate cyclase-activating polypeptide (PACAP) receptor, δ-opioid receptor (DOR), calcitonin gene-related peptide (CGRP) receptors are G-protein coupled receptors, whereas Acid-sensing ion channels (ASIC), transient receptor potential channels (TRP), and potassium channels (K_ATP_) are ion channels.

## Data Availability

Not applicable.

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
