# Peer review of "Novel Therapeutic Targets for Migraine"

_biomedicines, 2023, doi:10.3390/biomedicines11020569_

Round 1
Reviewer 1 Report
The authors present a review focusing on drugs under development for migraine, which are beyond CGRP(r) monoclonal antibodies, gepants and ditans.
The review is thorough, up-to-date and helps to navigate well in this area of drug development. In the face of a thorough review of Clinicaltrails.gov, it is easy to see that the introduction and conclusions lack reference papers that might better orient journal readers who are unfamiliar with migraine.
I therefore suggest that the following papers be considered for inclusion and discussion:
PMID: 35410119 PMID: 36471250 PMID: 36597043
PMID: 36180824 PMID: 35690723 PMID: 35690723
Author Response
Point 1: The authors present a review focusing on drugs under development for migraine, which are beyond CGRP(r) monoclonal antibodies, gepants and ditans.
Response 1: The authors appreciate your effort in reviewing the article. Thank you for your comments.
Point 2: The review is thorough, up-to-date and helps to navigate well in this area of drug development. In the face of a thorough review of Clinicaltrails.gov, it is easy to see that the introduction and conclusions lack reference papers that might better orient journal readers who are unfamiliar with migraine.
I therefore suggest that the following papers be considered for inclusion and discussion:
PMID: 35410119   
PMID: 36471250  
 PMID: 36597043
PMID: 36180824   
PMID: 35690723  
Response 2: Thank you for your comments and recommendations. The authors have included the suggested papers as applicable.
Reviewer 2 Report
It seemed to me that the possibilities that you considered dealt mainly with factors involved in pain production in migraine rather than factors involved in the initiating phases of migraine attacks, or in the vascular changes, though I can see that the approaches you suggest might offer useful benefits.
Also, I wondered if you assumed more familiarity with ideas about migraine pathogenesis than the average reader might have.
My only suggestion is that you consider adding a table explaining all the abbreviations that you use. I think a reader unfamiliar with the area might be overwhelmed by the difficulty attendant on having to repeatedly go back to find where each was first used (and then sometimes finding there was no explanation), and might abandon reading the paper, whereas if the abbreviations were consolidated in one place reader time and effort would be reduced.
Author Response
Point 1: It seemed to me that the possibilities that you considered dealt mainly with factors involved in pain production in migraine rather than factors involved in the initiating phases of migraine attacks, or in the vascular changes, though I can see that the approaches you suggest might offer useful benefits. 
Response 1: The authors appreciate your effort on reviewing the article. Thank you for your comments
Point 2: Also, I wondered if you assumed more familiarity with ideas about migraine pathogenesis than the average reader might have.
Response 2: As recommended the authors have added a brief pathophysiology of migraine to aid average readers in their understanding.
Point 3: My only suggestion is that you consider adding a table explaining all the abbreviations that you use. I think a reader unfamiliar with the area might be overwhelmed by the difficulty attendant on having to repeatedly go back to find where each was first used (and then sometimes finding there was no explanation), and might abandon reading the paper, whereas if the abbreviations were consolidated in one place reader time and effort would be reduced.
Response 3: As suggested by the reviewer, the authors have added a table listing all abbreviations used in the article for the ease of the reader.
Reviewer 3 Report
This is a well-written review concerning the new therapeutic target of migraine.
I would suggest modifying figure 1, adding the mechanisms downstream the receptors' activation
Author Response
Point 1: This is a well-written review concerning the new therapeutic target of migraine
Response 1: The authors appreciate your effort on reviewing the article. Thank you for your comments.
Point 2: I would suggest modifying figure 1, adding the mechanisms downstream the receptors' activation.
Response 2: Thank you for the suggestion. These various receptors likely elicit complex interacting pain signaling pathways which are beyond the scope of this narrative review.